# Corrosion Behavior and Comprehensive Evaluation of Al$_{0.8}$CrFeCoNiCu$_{0.5}$B$_{0.1}$ High-Entropy Alloy in 3.5% NaCl Solution

**Yanzhou Li** [1,2], **Yan Shi** [3,4,*], **Rongna Chen** [5], **Hua Lin** [1,2] and **Xiaohu Ji** [1,2]

1 School of Mechanical and Vehicle Engineering, West Anhui University, No.1 Yueliangdao Road, Hanhui, Lu'an 237010, China; 04000152@wxc.edu.cn (Y.L.); linhua@wxc.edu.cn (H.L.); 04000169@wxc.edu.cn (X.J.)
2 Innovation Platform of High-Performance Complex Manufacturing Intelligent Decision and Control, West Anhui University, Lu'an 237012, China
3 School of Electromechanical Engineering, University of Science and Technology, No. 7089 Weixing Road, Changchun 130022, China
4 National Base of International Science and Technology Cooperation in Optics, No. 7089 Weixing Road, Changchun 130022, China
5 Luan Hengyuan Machinery Co., Ltd., Lu'an 237012, China; hzd3276032@163.com
* Correspondence: shiyan@cust.edu.cn

**Abstract:** In this study, Al$_{0.8}$CrFeCoNiCu$_{0.5}$B$_{0.1}$ high-entropy alloy coating was prepared on the surface of 5083 aluminum alloy using laser cladding technology. The corrosion behavior of the coating and substrate in 3.5% NaCl solution was analyzed using experimental methods, including polarization curves and electrochemical impedance spectroscopy. The corrosion current density of Al$_{0.8}$CrFeCoNiCu$_{0.5}$B$_{0.1}$ coating is 2.04 × 10$^{-7}$ A/cm$^2$. The passivation range width reaches 2.771 V, and these polarization test results are superior to the substrate. The Al$_{0.8}$CrFeCoNiCu$_{0.5}$B$_{0.1}$ coating exhibited selective corrosion behavior, with the Cu-rich FCC1 phase and Cr-poor phase being susceptible to corrosion, leading to localized pitting and intergranular corrosion traces, but the corrosion did not spread extensively. The intergranular distribution of Cu is the main reason for the intergranular corrosion trace features. In contrast, the substrate exhibited overall corrosion. The Nyquist plot of the Al$_{0.8}$CrFeCoNiCu$_{0.5}$B$_{0.1}$ coating consisted of a single capacitive semicircle arc in the high-frequency region with a larger radius than the substrate. In conclusion, using the Al$_{0.8}$CrFeCoNiCu$_{0.5}$B$_{0.1}$ high-entropy alloy as a coating can significantly improve the corrosion resistance of the 5083 aluminum alloy substrate.

**Keywords:** high-entropy alloy; laser cladding; corrosion resistance; corrosion behavior

## 1. Introduction

Aluminum has become the material of choice for lightweight designs because of its high specific strength, specific stiffness, and low-cost properties [1–3]. However, its low surface hardness and lack of wear and corrosion resistance limit its application in specific fields. Popular surface-strengthening methods for aluminum alloys include chemical plating, electrodeposition techniques, and laser cladding. Among these, laser cladding stands out due to its lower pollution levels, high work efficiency, and the ability to apply thick coatings, advantages that make it superior to other surface treatment methods [4–8]. For instance, Hwang et al. [9] utilized anodizing technology to generate an anodic oxide film on the surface of the 5083 aluminum alloy. By conducting electrochemical corrosion tests in natural seawater, they discovered that the corrosion current density of the anodized sample was approximately 4.2 times lower than that of the base material, demonstrating its superior corrosion resistance. However, the thickness of the oxide film was only 16.8 um, indicating a certain limitation of the anodizing method in applications that require thicker coatings.

Although an efficient and environmentally friendly technology, laser cladding has been widely used in material surface modification [10,11]. Due to the strong electronegativity of aluminum, complex intermetallic compounds are easily formed after traditional materials react with aluminum in the molten pool, resulting in poor coating quality.

As a new alloy material, high-entropy alloy (HEA) was proposed by Yeh et al. in 2004 and is defined as an alloy system consisting of 5 to 13 isotonic ratios or near-isotonic ratios [12]. Compared with conventional alloys, HEAs have a high-entropy effect, hysteresis diffusion effect, and lattice distortion effect, which enable them to inhibit the formation of ordered intermetallic compounds during solidification and tend to form simple solid solution structures [5,13–18]. Based on the four effects of HEAs, using them as coatings for laser cladding of aluminum alloy surfaces is beneficial to suppress the problems of poor coating quality caused by the dilution behavior of aluminum and, at the same time, improve the coating performance.

The Al-Cr-Fe-Co-Ni-Cu system is a typical HEA of the transition group. Scholars have investigated its organization and properties. TONG et al. [19] prepared $Al_xCoCrCuFeNi$ (x = 0~3.0) HEA by arc melting, and the alloy exhibited a simple fcc/bcc structure. The alloy has a single fcc solid solution structure at low aluminum content. At x = 0.8, the alloy shows a mixed fcc eutectic phase and bcc eutectic structure. When x > 1.0, the alloy undergoes Spinodal decomposition, forming a modulated plate structure. When x > 2.8, the alloy obtains a single-phase ordered bcc structure. The high-entropy effect and hysteresis diffusion effect synergistically promoted the formation of l simple phases, submicron, and nanoprecipitates in the alloy. Zhou et al. [20] found that the cast $Al_{0.4}CoCrCuFeNi$ high-entropy alloy showed FCC1 and FCC2 phases. The alloy is antibacterial in corrosive environments containing Gram-negative Pseudomonas aeruginosa and Gram-positive Bacillus vietnamensis. Its mechanical properties are better than those of 304 stainless steel. Chen et al. [21] found that increasing Cu content reduces the corrosion resistance and passivation film protection of $Al_{0.3}Cu_xCoCrFeNi$ high-entropy alloy, but passivation in 30 wt% $HNO_3$ and 1 wt% NaOH improves the $Cr_3^+$ content of the passivation film and reduces the pitting tendency. Excessive Cu content will promote the alloy containing Al and Cu-rich phases to compete, weakening the passivation treatment effect and destroying the uniformity of the passivation film, leading to severe galvanic coupling corrosion. In the study conducted by Günen et al. [22], laser-clad AlCrFeCoNi high-entropy alloy (HEA) coatings were prepared on the surface of AISI 316L stainless steel. After a 4 h pack boriding treatment at 1000 °C, a complex boride layer consisting of $(CoFe)B_2$, $(CrFe)B_2$, and $Cr_2Ni_3B_6$ phases formed on the surface of the HEA coating. The boriding process increased the hardness of the coating from an initial 6.14 ± 2.06 GPa to a range of 15.95 ± 0.7 to 20.15 ± 4.50 GPa. While the greatest boride layer thickness was achieved in the sample borided in vacuum, the highest surface hardness was obtained in the sample borided in air. The borided coatings demonstrated better wear resistance and lower friction coefficients compared to untreated samples, both at 25 °C and 650 °C. However, the wear losses at 650 °C significantly exceeded those at 25 °C.

Several scholars have delved into the transformation of the microstructure in high-entropy alloys (HEAs) produced via laser cladding. For instance, Liu et al. [23] fabricated a dual-layer $CoCrFeMnTi_{0.2}$ HEA coating on 15CrMn steel using this technique. They observed a distinctive shift in the microstructure. The first layer mainly consisted of FCC solid solution equiaxed crystals and intergranular Laves phase precipitates. This structure evolved in the second layer, morphing into an FCC solid solution/Laves lamellar eutectic structure, surrounded by martensite and residual FCC solid solution phase. They attributed these transformations to the dilution effect from the dual-layer deposition process and the quenching effect induced by multi-track overlapping. Similarly, Li's team [24] employed laser cladding to produce double-layer $Al_{0.5}CoCrFeNiSi_{0.5}Ti_x$ coatings. They noted that an increase in the Ti content resulted in substantial changes in the coating's microstructure. The FCC solid solution in the first layer transformed from a cellular structure to a columnar dendritic one. In contrast, the second layer manifested a BCC solid solution, composed of

disordered BCC and ordered B2 phases. Additionally, the incorporation of Ti instigated the in situ synthesis of TiN and $TiSi_2$, considerably improving the coatings' hardness.

In the study of $Al_{0.8}CrFeCoNiCu_{0.5}B_x$ (x = 0, 0.1, 0.2, 0.3, 0.4) HEA coating by laser cladding on aluminum, our group [25] found that when the B content is less than 0.2, the coating presents FCC, BCC1, BCC2 simple phase structure, and M2B (M = Fe, Cr) phase will appear when the B coating continues to be added. As a result of M2B formation, the coating's hardness and wear resistance increase with the rising B content. The mechanical properties of $Al_{0.5}CrFeCoNiCu_{0.5}B_x$ coating are significantly better than the substrate. It is worth noting that the hardness of $Al_{0.8}CrFeCoNiCu_{0.5}B_{0.1}$ is 549HV0.2, approximately seven times that of the substrate, and its wear rate is $5.53 \times 10^{-6}$ $mm^3/Nm$, which is two orders of magnitude less than that of the substrate. However, when the B content is more significant than 0.1, the bond strength of $Al_{0.5}CrFeCoNiCu_{0.5}B_x$ coating and the substrate is lower, and the shear fracture is brittle. Based on the above research, this study focuses on the $Al_{0.8}CrFeCoNiCu_{0.5}B_{0.1}$ coating, which exhibits high bonding strength to the substrate. Utilizing polarization tests, EIS, and SEM techniques, the corrosion resistance behavior of the $Al_{0.8}CrFeCoNiCu_{0.5}B_{0.1}$ coating in a 3.5% NaCl solution is investigated. The feasibility of $Al_{0.8}CrFeCoNiCu_{0.5}B_{0.1}$ coating to improve the corrosion resistance of aluminum alloy surface is given.

## 2. Experimental Details

The experimental base material is a $50 \times 30 \times 10$ mm 5083 aluminum alloy, with the oxide film removed mechanically before cladding. Cr, Fe, Co, Ni, B, and Cu powders are chosen and manufactured using either atomization or reduction techniques, ensuring purities of over 99.5% and particle sizes ranging from 200 to 325 mesh. The composition ratios can be found in Table 1. After mixing, they were homogenized by 15 h of ball milling, using a QM–QX planetary ball mill for dry grinding. To guarantee the cladding layer's quality, the substrate was preheated to 200 °C, and high-purity argon gas served as the protective gas for coaxial powder feeding at a rate of 5 L/min. The laser cladding process parameters were a laser power of 1500 W, a spot diameter of 2.5 mm, a scanning speed of 5 mm/s, and a 3 g/min powder feeding speed. After the coating preparation, it was wrapped in insulation cotton and slowly cooled to prevent the formation of cracks.

**Table 1.** Nominal component of $Al_{0.8}FeCoNiCrCu_{0.5}B_{0.1}$ (at%).

| Alloy | Al | Cr | Fe | Co | Ni | Cu | B |
|---|---|---|---|---|---|---|---|
| $Al_{0.8}CrFeCoNiCuB_{0.1}$ | 14.8148 | 18.5185 | 18.5185 | 18.5185 | 18.5185 | 9.2593 | 1.8519 |

The electrochemical sample preparation method involved welding copper wires to the sample and encapsulating it with epoxy resin, exposing a 10 mm × 10 mm test surface. The sample was polished step by step with SiC sandpaper up to 1500 grit, then cleaned with deionized water and anhydrous ethanol and dried with cold air. Electrochemical tests were performed on a Zennuim pro electrochemical workstation using a traditional three-electrode system, with a saturated calomel electrode (SCE) as the reference electrode, a Pt electrode as the counter, and $Al_{0.8}CrFeCoNiCu_{0.5}B_{0.1}$ HEA as the working electrode. The functional test area of the working electrode was 1 $cm^2$. Corrosion tests were conducted in a 3.5% NaCl solution and simulated seawater at room temperature. Following the corrosion test, the surface morphology of the samples was observed using an SEM. Before the electrochemical test, the sample was immersed in the test solution for 1 h to ensure a relatively stable surface state. The sample's open circuit potential (OCP) was initially measured and allowed to stabilize before performing the EIS test. The EIS test frequency range was from 100 mHz to 10 kHz, with an oscillating potential of 10 mV. EIS data were fitted and analyzed using Zview software. The potentiodynamic polarization curve test's initial potential was −1.5 V, and the final potential was 1.5 V.

The electrochemical sample preparation involved affixing copper wires to the sample and encapsulating the entirety in epoxy resin, thus leaving an exposed 10 mm × 10 mm

test surface. This sample was then systematically polished using SiC sandpaper, gradually advancing to a 1500 grit finish. The sample was then meticulously cleaned with deionized water, anhydrous ethanol, and dried using cold air. The electrochemical tests were executed on a Zennium Pro electrochemical workstation utilizing a conventional three-electrode system. This system included a saturated calomel electrode (SCE) as the reference electrode, a platinum electrode as the counter, and $Al_{0.8}CrFeCoNiCu_{0.5}B_{0.1}$ HEA as the working electrode. Corrosion tests were conducted in both a 3.5% NaCl solution and simulated seawater at ambient room temperature. Post-corrosion test, the surface morphology of the samples was examined using scanning electron microscopy (SEM). Prior to the electrochemical tests, the sample was submerged in the test solution for an hour, ensuring a relatively stable surface state. The open circuit potential (OCP) of the sample was initially measured and given time to stabilize before performing the EIS test. The frequency range for the EIS test was set between 100 mHz and 10 kHz, with an oscillating potential of 10 mV. EIS data were fitted and analyzed using Zview software. The potentiodynamic polarization curve test commenced at an initial potential of −1.5 V and ended at a final potential of 1.5 V.

## 3. Results and Discussion

### 3.1. HEA Coating and Substrate Polarization Curve

Figure 1 shows the polarization curves of the substrate and $Al_{0.8}CrFeCoNiCu_{0.5}B_{0.1}$ coating in a 3.5% NaCl solution at room temperature. The polarization curves of the coating and the substrate consist of four parts: the cathodic zone, the anodic dissolution zone, the passive zone, and the transpassive zone. In the cathodic zone, the current density decreases as the potential increases, with the cathodic reaction corresponding to the oxygen evolution process [26]. As the voltage increases, the sample undergoes an anodic dissolution process, causing a sharp rise in the self-corrosion current density [27]. As the voltage continues to increase, the self-corrosion current density gradually stabilizes, entering the passive zone. With a further increase in voltage, the self-corrosion current density increases again, entering the transpassive zone. The anodic portion reflects essential characteristics related to the material's corrosion resistance.

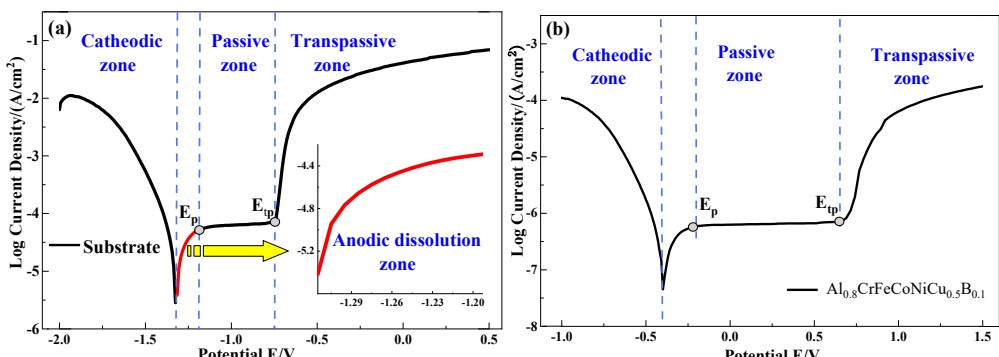

**Figure 1.** The polarization curves of the substrate and $Al_{0.8}CrFeCoNiCu_{0.5}B_{0.1}$ HEA coating. (**a**) the polarization curve of the substrate, with the inset showing an enlarged view of the anodic dissolution area on the curve; (**b**) the polarization curve of the coating.

The corrosion current density can represent the metal's corrosion rate. The higher the Icorr, the greater the corrosion rate of the metal [28]. By calculating the electrochemical parameters of the samples using the Tafel extrapolation method, the Icorr of the $Al_{0.8}CrFeCoNiCu_{0.5}B_{0.1}$ coating is found to be $2.04 \times 10^{-7}$ A/cm$^2$, which is significantly lower than that of the 5083 aluminum alloy; the results are shown in Table 2. This demonstrates that the corrosion rate of the $Al_{0.8}CrFeCoNiCu_{0.5}B_{0.1}$ coating is much lower than that of the substrate.

**Table 2.** The electrochemical parameters of $Al_{0.8}CrFeCoNiCu_{0.5}B_{0.1}$ HEA coatings and substrate.

| Solution | Samples | $E_{corr}$/V | $I_{corr}$/(A/cm$^2$) | Ep/V | Etp/V | $\Delta$E/V |
|:---:|:---:|:---:|:---:|:---:|:---:|:---:|
| 3.5% NaCl | Substrate | −1.32 | $4.12 \times 10^{-5}$ | −1.182 | −0.740 | 0.442 |
| | $B_{0.1}$ | −0.415 | $2.04 \times 10^{-7}$ | −2.122 | 0.649 | 2.771 |

Each sample displays a passivation zone, indicating that as the electrochemical corrosion process progresses, a passivation layer forms on the electrode surface, effectively reducing the erosion of the working electrode surface by Cl ions during the electrochemical corrosion process. During anodic polarization, the potential at the onset of passivation is the passivation potential (Ep), while the potential when the passive film starts to break is the transpassive potential (Etp). The difference between the transpassive potential and the passivation potential ($\Delta$E) reflects the stability of the passive film. Table 2 demonstrates that the passive region width of the $Al_{0.8}CrFeCoNiCu_{0.5}B_{0.1}$ coating is significantly greater than that of the substrate, suggesting that the passive film created by the HEA coating provides enhanced protection. In a corrosive solution environment, Cl ions adsorb onto the surface of the passivation film and then gradually migrate toward the interior of the electrode. A dense passive film can act as a barrier to block Cl ions' penetration, protecting the electrode. The passive film's protective effect in the $Al_{0.8}CrFeCoNiCu_{0.5}B_{0.1}$ coating surpasses that of the substrate, likely attributable to the corrosion-resistant properties of the element Cr. In the coating, the self-corrosion current density is lower than that of the substrate, indicating better corrosion resistance. To further investigate the corrosion behavior of the $Al_{0.8}CrFeCoNiCu_{0.5}B_{0.1}$ HEA coating, the corroded morphology is examined using SEM, as shown in Figure 2.

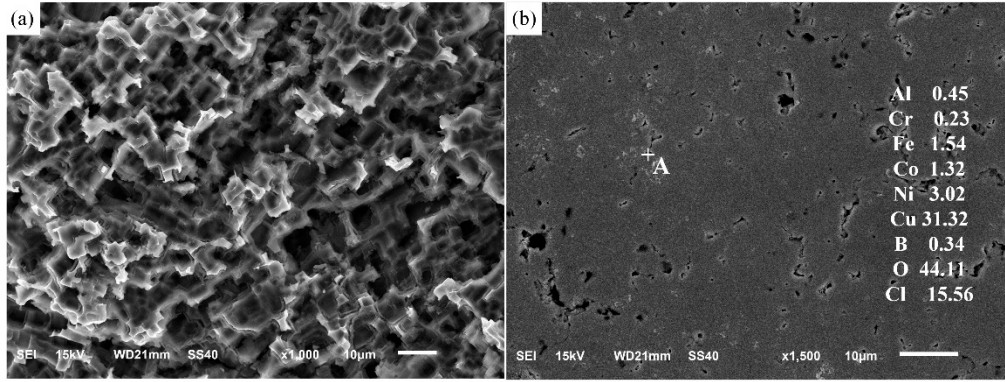

**Figure 2.** Corrosion morphology of substrate and the $Al_{0.8}CrFeCoNiCu_{0.5}B_{0.1}$ HEA coating. (**a**) the corrosion morphology of the substrate; (**b**) the corrosion morphology of the coating, with spectral information observable within the figure.

*3.2. Corrosion Behavior of Coatings and Substrates*

Figure 2 showcases the microscopic corrosion morphologies of the 5083 aluminum alloy and the $Al_{0.8}CrFeCoNiCu_{0.5}B_{0.1}$ high-entropy alloy (HEA) coating. From previous studies on the microstructure of the 5083 aluminum alloy [29], it was found that the alloy consists of an Al matrix phase, an Al(Mn, Fe, Cr) phase presenting as fish-bone-like structures, and a $Mg_2Si$ phase dispersed as blocky particles throughout the matrix. After conducting polarization tests, it was identified that the 5083 alloy substrate had been extensively corroded, as seen in Figure 2a. In contrast, the surface of the HEA coating maintained its integrity, although corrosion products were evident. Figure 2b shows that the $Al_{0.8}CrFeCoNiCu_{0.5}B_{0.1}$ coating exhibits selective corrosion characteristics, with some pitting irregularly distributed. However, some corrosion pits also display intergranular corrosion traces, yet these pits do not propagate further. EDS analysis of the corrosion products in the coatings reveals that the products are primarily oxides and chlorides of Cu.

Figure 3 shows the polarization testing mechanism of $Al_{0.8}CrFeCoNiCu_{0.5}B_{0.1}$ coating. Our previous studies have revealed that the $Al_{0.8}CrFeCoNiCu_{0.5}B_{0.1}$ coating is primarily

composed of FCC1, BCC1, and BCC2 phases [25]. The FCC1 phase, which is predominantly found within the intergranular structure, is rich in Cu. The BCC1 phase is rich in Fe and Cr, while the BCC2 phase is rich in Al and Ni. Boron is present within the structure as a solid-solution element. This contrasts with the findings of G ü nen et al., where boronizing treatment of AlCoCrFeNi high-entropy alloy led to the formation of $(CoFe)B_2$, $(CrFe)B_2$, and $Cr_2Ni_3B_6$ phases under different conditions. Given that no second phase containing B was detected in the XRD test, it can be inferred that B exists in the high-entropy alloy matrix as a solid-solution element. Combined with the EDS analysis of the corrosion products, it is confirmed that the Cu-rich FCC1 phase is preferentially corroded.

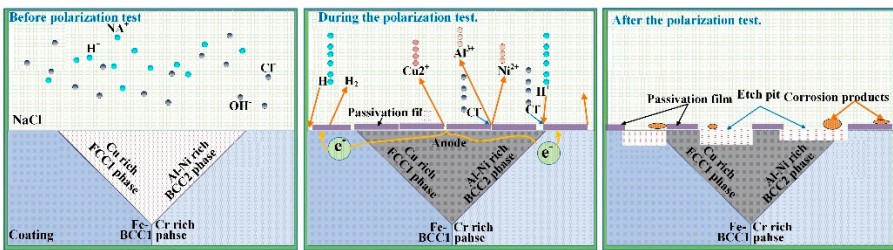

**Figure 3.** Polarization testing mechanism diagram of $Al_{0.8}CrFeCoNiCu_{0.5}B_{0.1}$ coating.

Regarding passivation ability, the passivation ability of Cr, Ni, Co, and Cu in solutions containing $Cl^-$ decreases in that order [30]. As a result, Cr-depleted phases are more susceptible to preferential attack. It can be inferred that during the polarization process, $Cl^-$ first adsorbs onto weak areas of the passive film. Since the second phase forms micro-galvanic couples with the surrounding media, the electrochemical reactions in this area are more intense. As the matrix is a Cr-depleted phase, pitting initiates on the matrix surface. The formation of pitting is accompanied by the generation of a passive film; however, the presence of the second phase disrupts the film's continuity. Therefore, as the electrochemical corrosion progresses, pitting is more likely to penetrate the passive film and extend along the matrix phase, leading to localized active dissolution corrosion centered around the initial pitting in the coating. Consequently, the Cr-depleted phase is more prone to pitting.

From the perspective of cathodic reactions, the Cl ion content in the electrolyte is relatively high, leading to oxygen reduction reactions at the cathode, reducing the oxygen concentration within the corrosion pits. Meanwhile, the oxygen concentration outside the pits is higher, forming an oxygen concentration cell. As the reaction continues, the concentration of metal ions within the corrosion pits gradually increases. Under the electric field generated by the oxygen concentration cell, Cl ions in the electrolyte continuously migrate and accumulate within the pits. The highly chlorinated solution in the corrosion pits has high conductivity, resulting in a low internal resistance value for the occluded cell. The solubility of oxygen in the pit solution is low, and oxygen diffusion is relatively difficult, causing the local oxygen supply of the occluded cell to be significantly limited. These factors hinder the passivation of metal elements within the corrosion pits, causing the metal elements to remain in an active state. The deposition layer formed by the corrosion products obstructs ion diffusion and convection at the pit opening, preventing the pit solution from being diluted. These factors collectively promote the formation of the oxygen concentration cell effect, leading to a greater extent of dissolution of the metal elements within the corrosion pits and the localized appearance of corrosion pits.

The coating's corrosion morphology exhibits intergranular corrosion traces, which are related to the distribution of Cu along grain boundaries. In the electrolyte solution, the galvanic couple's cathode consists of the Fe- and Cr-rich BCC1 phase, whereas the anode consists of the Cu-rich FCC1 phase and the Al- and Ni-rich BCC2 phase. The corrosion of the high-entropy alloy in NaCl solution is mainly achieved through chloride ions, which lower the redox potential of the various elements in the alloy. As the anodes, the Cu-rich

and Al-, Ni-rich phases undergo active dissolution due to the loss of electrons, resulting in preferential erosion by corrosive ions at the coating's grain boundaries.

The dissolution process of the anode involves the metal entering the corrosive solution in ionic form while electrons remain in the metal matrix. The equation for this process is as follows:

$$Al + 2H_2O + Cl^- \rightarrow Al^{3+} \cdot 2H_2O + Cl + 3e^-$$

$$Cu + 2H_2O + Cl^- \rightarrow Cu^{2+} \cdot 2H_2O + Cl + 2e^-$$

$$Ni + 2H_2O + Cl^- \rightarrow Ni^{2+} \cdot 2H_2O + Cl + 2e^-$$

On the surface of the cathodic region, the residual electrons on the metal matrix surface react with adsorbed oxygen and water to form hydroxide ions. The reaction equation is as follows:

$$O_2 + 4e^- + 2H_2O \rightarrow 4OH^-$$

Upon dissolution of NaCl, it reacts with the metal ions above to form corrosion products. The equation is as follows:

$$Al^{3+} \cdot 2H_2O + 3Cl^- \rightarrow Al_3Cl \cdot 2H_2O$$

$$Al^{3+} + 3OH^- \rightarrow Al(OH)_3$$

$$Cu^{2+} \cdot 2H_2O + 3Cl^- \rightarrow CuCl_2 \cdot 2H_2O$$

$$Cu^{2+} + 2OH^- \rightarrow Cu(OH)_2$$

$$Ni^{2+} \cdot 2H_2O + 3Cl^- \rightarrow NiCl_2 \cdot 2H_2O$$

$$Ni^{2+} + 2OH^- \rightarrow Ni(OH)_2$$

### 3.3. Impedance Spectrum Testing of Coatings and Substrates

Figure 4 shows the Nyquist plots of $Al_{0.8}CrFeCoNiCu_{0.5}B_{0.1}$ HEA coating and the substrate in 3.5% NaCl solution at open circuit potential. The Nyquist plot of the coating consists of a single capacitive semicircle in the high-frequency range, with the center below the $X$-axis, indicating charge transfer between the working electrode surface and the solution. The impedance arc radius of the $Al_{0.8}CrFeCoNiCu_{0.5}B_{0.1}$ HEA coating is more significant than that of the substrate, indicating that its corrosion resistance is higher than that of the substrate but lower than that of $Al_{0.8}CrFeCoNiCu_{0.5}$ coating, which is consistent with the polarization test results.

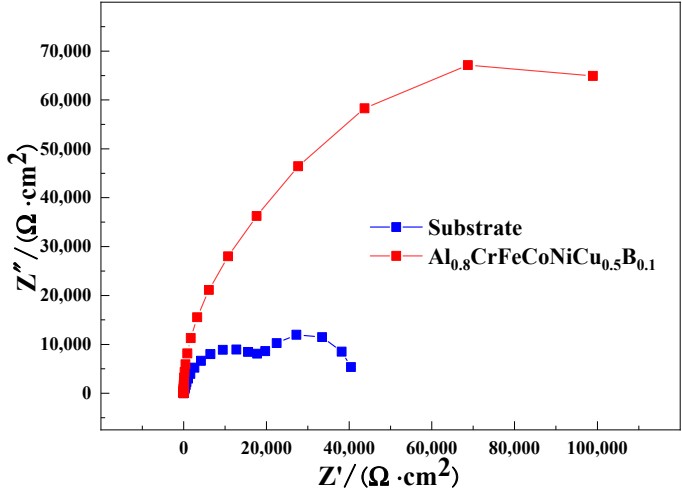

**Figure 4.** Nyquist plots of the $Al_{0.8}CrFeCoNiCu_{0.5}B_{0.1}$ HEA coating and substrate.

Figure 5 shows the equivalent circuit diagram of the $Al_{0.8}CrFeCoNiCu_{0.5}B_{0.1}$ coating for EIS. The equivalent circuit includes solution resistance, charge transfer resistance, and double-layer capacitance at the interface between the electrode and the solution. The formation of the double-layer capacitance is due to the following reasons. During the electrochemical reaction process, the electrolyte solution promotes the departure of metal from the working electrode surface in the form of ions, leaving the metal with a negative potential due to the loss of positive ions. Metal ions in the solution are attracted and re-adsorbed onto the metal surface by anions or electrons on the metal electrode surface. The anions ($Cl^-$, $OH^-$) in the electrolyte solution are attracted by the cations on the metal surface, gather around them, and form ionic precipitates through chemical reactions. When the concentration of the solution reaches a specific value, the dissolution of the metal and the precipitation of ions reach a dynamic equilibrium, forming a double layer at the interface between the metal and the solution, namely the compact layer and the diffusion layer. Considering that the electrode surface is in a non-uniform and smooth state, the double-layer capacitance of the electrode during EIS testing responds differently to time and frequency compared to solid-state capacitance [31]. Replacing the double-layer capacitance with pure capacitance will cause a dispersion effect, leading to deviations from the results in impedance spectroscopy measurements. Therefore, a constant phase element (CPE1) is used to replace the equivalent circuit parameter element, and its impedance $Z_{CPE}$ calculation formula is [32]:

$$Z_{CPE} = Y_0{}^{-1}(j\omega)^{-n} = Y_0\omega^{-n}\cos\frac{n\pi}{2} + jY_0\omega^n\sin\frac{n\pi}{2}$$

a is the proportionality factor, b is the imaginary number, c is the angular frequency of the test current, d is the dispersion coefficient, and e represents the degree of dispersion of the equivalent circuit capacitance, that is, the degree of non-uniform current distribution, with calculated values ranging from $-1$ to 1. When $n = 1$, it represents an ideal capacitive circuit; when $n = 0.7$, it means a porous electrode circuit; when $n = 0.5$, it represents Warburg diffusion impedance; when $n = 0$, it represents a purely resistive circuit; when $n = -1$, it represents a purely inductive circuit [33]. The closer the value of $n$ is to 1, the more uniform and dense the electrode surface, and the further it deviates from 1, the more the electrode surface is influenced by the porous structure. Table 3 shows that the electrode Rs of the coating solution obtained smaller values, indicating that they had good contact with the electrode in a 3.5% NaCl solution. The CPE1-$n$ value is between 0.5 and 1, meaning a high surface density of the coating, making it less prone to comprehensive corrosion in a 3.5% NaCl solution.

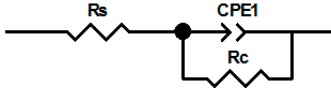

**Figure 5.** The equivalent circuit diagram of the $Al_{0.8}CrFeCoNiCu_{0.5}B_{0.1}$ coating.

**Table 3.** Equivalent circuit parameters of the $Al_{0.8}CrFeCoNiCu_{0.5}B_{0.1}$ coating.

| Alloy | $R_s$ ($\Omega \cdot cm^2$) | CPE1 | | $R_c/k\Omega \cdot cm^2$ |
| | | Y ($uF \cdot cm^2$) | $n$ | |
|---|---|---|---|---|
| $Al_{0.8}CrFeCoNiCu_{0.5}B_{0.1}$ | 3.0 | 18.5 | 0.897 | 201.3 |

### 3.4. Comprehensive Performance of Coating

To comprehensively evaluate the feasibility of using $Al_{0.8}CrFeCoNiCu_{0.5}B_{0.1}$ for surface modification of aluminum alloys, the overall performance of the coating has been analyzed [25]. Figure 6 shows the hardness and wear rate of the $Al_{0.8}CrFeCoNiCu_{0.5}B_{0.1}$ high-entropy alloy coating compared to the base material. The hardness of the coating reached 549HV0.2, which is eight times that of the base material, and the wear rate was

$5.53 \times 10^{-6}$ mm$^3$/Nm, two orders of magnitude lower than the base material. Observations of the worn surfaces revealed serious layer cracking and plastic deformation in the base material. In contrast, the worn surface of the Al$_{0.8}$CrFeCoNiCu$_{0.5}$B$_{0.1}$ coating was smoother, with visible grooves and wear particles, without any delamination or plastic deformation, suggesting that the wear mechanism is adhesive wear and abrasive wear. Therefore, in terms of corrosion resistance, hardness, and wear resistance, this coating outperforms the base material on all fronts. From the perspective of material performance, Al$_{0.8}$CrFeCoNiCu$_{0.5}$B$_{0.1}$ demonstrates significant feasibility as a surface modification coating for aluminum alloys.

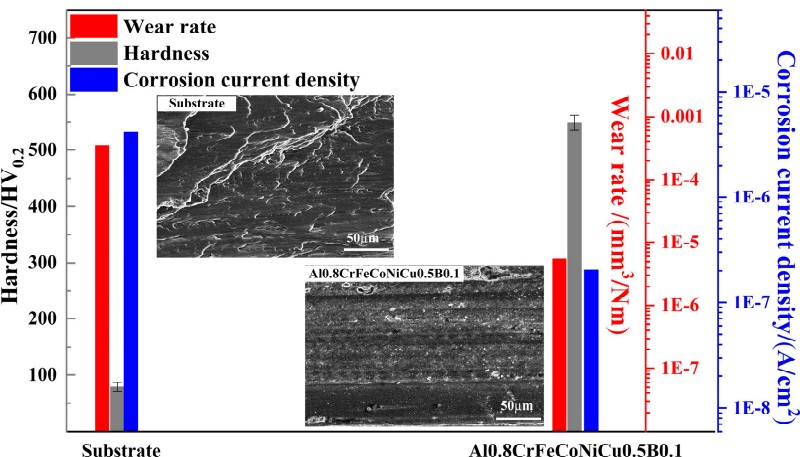

**Figure 6.** Comprehensive properties of Al$_{0.8}$CrFeCoNiCu$_{0.5}$B$_{0.1}$ coating and substrate.

## 4. Conclusions

1. The corrosion current density of the Al$_{0.8}$CrFeCoNiCu$_{0.5}$B$_{0.1}$ coating is $2.04 \times 10^{-7}$ A/cm$^2$, significantly lower than the 5083 aluminum alloy. The passivation range width of the coating is 2.771 V, wider than the substrate.

2. The Al$_{0.8}$CrFeCoNiCu$_{0.5}$B$_{0.1}$ coating exhibits selective corrosion behavior, with the Cu-rich FCC1 phase and Cr-poor phase being prone to attack, resulting in partial pitting and intergranular corrosion traces, but the corrosion does not spread extensively. The intergranular distribution of Cu is the main reason for the intergranular corrosion trace features. The 5083 base material undergoes overall corrosion.

3. The Nyquist plot of the Al$_{0.8}$CrFeCoNiCu$_{0.5}$B$_{0.1}$ coating consists of a single capacitive semicircle arc in the high-frequency region with a larger radius than the base material. The coating forms a double-layer capacitance in the solution, making it difficult for overall corrosion to occur in a 3.5% NaCl solution.

The present study demonstrates that, compared to 5083 aluminum alloy, Al$_{0.8}$CrFeCoNiCu$_{0.5}$B$_{0.1}$ high-entropy alloy exhibits superior corrosion resistance. When used as a coating for aluminum alloys, it significantly expands the application possibilities of aluminum alloys, such as in marine ship hull applications. This improved corrosion resistance is primarily attributed to the passivation layer formed by the HEA coating, which effectively withstands the corrosive marine environment. Given the lightweight advantages of aluminum alloys, coupled with the high corrosion resistance brought by the HEA coating, there is a potential to provide a broader application platform for aluminum alloys, which are low-cost lightweight alloys. As laser cladding technology matures and the cost of high-entropy alloys decreases in the future, high-entropy alloys are expected to become excellent materials for laser cladding on the surface of aluminum alloys.

**Author Contributions:** Writing—review & editing, Y.L.; Visualization, X.J.; Supervision, H.L.; Project administration, Y.S.; Funding acquisition, Y.S. and R.C. All authors have read and agreed to the published version of the manuscript.

**Funding:** This work was financially supported by the Natural Science Research Project of Anhui Provincial Department of Education (Grant Nos. KJ2021A0947, KJ2020A0626 and KJ2020A0625), University level natural science research project of West Anhui University (Grant Nos. WXZR202116, WGKQ2021068), the High-level Talents Research Project of West Anhui University (Grant No. WGKQ2021068, WGKQ 201802004), and the Key Research and Development Project of Anhui Province (Grant No. 202104a06020004).

**Data Availability Statement:** No new data were created or analyzed in this study. Data sharing is not applicable to this article.

**Conflicts of Interest:** The authors declare no conflict of interest.

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
