# Peer review of "Corrosion Behavior and Comprehensive Evaluation of Al0.8CrFeCoNiCu0.5B0.1 High-Entropy Alloy in 3.5% NaCl Solution"

_lubricants, doi:10.3390/lubricants11070282_

Round 1

Reviewer 1 Report

The corrosion resistance of Al0.8CrFeCoNiCu0.5B0.1 HEAs deposited on Al5083 by laser-cald method was investigated. The results are important in showing that laser-clad HEAs provide better corrosion resistance than Al5083. But some shortcomigs need to be taken care of in the article. These matters are given below.

1- It should be discussed that the addition of B to laser clad HEA alloys can be alloyed or applied in coatings as a secondary process. https://doi.org/10.1016/j.surfcoat.2022.128830.

2- Indicate which standard the corrosion tests were based on.

3- Microstructure pictures and EDS analyzes of laser-clad HEAs obtained in the result and discsusion section should be given.

4- In addition, the XRD phase structure of laser-clad HEAs should be presented so that it should be determined that the FCC1 or FCC2 structure is formed.

5- The reason for the improving in corrosion resistance should be examined in depth by making use of the literature in the discussion section.

Author Response

Response to Reviewer 1 Comments

Point 1: It should be discussed that the addition of B to laser clad HEA alloys can be alloyed or applied in coatings as a secondary process. https://doi.org/10.1016/j.surfcoat.2022.128830.

Response 1: The introduction of the manuscript has been added, and the content is as follows:

In the study conducted by Günen et al., laser-clad AlCrFeCoNi high entropy alloy (HEA) coatings were prepared on the surface of AISI 316L stainless steel. After a 4-hour pack boriding treatment at 1000°C, a complex boride layer consisting of (CoFe)B2, (CrFe)B2, and Cr2Ni3B6 phases formed on the surface of the HEA coating. The boriding process increased the hardness of the coating from an initial 6.14 ± 2.06 GPa to a range of 15.95 ± 0.7 to 20.15 ± 4.50 GPa. While the greatest boride layer thickness was achieved in the sample borided in vacuum, the highest surface hardness was obtained in the sample borided in air. The borided coatings demonstrated better wear resistance and lower friction coefficients compared to untreated samples, both at 25°C and 650°C. However, the wear losses at 650°C significantly exceeded those at 25°C.

In the discussion section 3.2, we have added the following content:

This contrasts with the findings of G ü nen et al., where boronizing treatment of AlCoCrFeNi high-entropy alloy led to the formation of (CoFe) B2, (CrFe) B2, and Cr2Ni3B6 phases under different conditions. Given that no second phase containing B was detected in the XRD test, it can be inferred that B exists in the high-entropy alloy matrix as a solid solution element.

Point 2: Indicate which standard the corrosion tests were based on.

Response 2:

During the polarization curve and impedance spectrum tests, we employed methodologies aligned with those found in certain academic papers. This approach ensures our experimental method aligns with established studies. The referenced papers include:

https://doi.org/10.1016/j.intermet.2021.107167. https://doi.org/10.3390/e22070740.

To further enhance the precision of our tests, we refined our experimental procedures as follows:

The electrochemical sample preparation involved affixing copper wires to the sample and encapsulating the entirety in epoxy resin, thus leaving an exposed 10mm x 10mm test surface. This sample was then systematically polished using SiC sandpaper, gradually advancing to a 1500 grit finish. The sample was then meticulously cleaned with deionized water, anhydrous ethanol, and dried using cold air.The electrochemi-cal tests were executed on a Zennium Pro electrochemical workstation utilizing a conventional three-electrode system. This system included a saturated calomel elec-trode (SCE) as the reference electrode, a platinum electrode as the counter, and Al0.8CrFeCoNiCu0.5B0.1 HEA as the working electrode. Corrosion tests were con-ducted in both a 3.5% NaCl solution and simulated seawater at ambient room tem-perature. Post-corrosion test, the surface morphology of the samples was examined using scanning electron microscopy (SEM).Prior to the electrochemical tests, the sample was submerged in the test solution for an hour, ensuring a relatively stable surface state. The open circuit potential (OCP) of the sample was initially measured and given time to stabilize before performing the EIS test. The frequency range for the EIS test was set between 100 mHz and 10k Hz, with an oscillating potential of 10 mV.EIS data were fit and analyzed using Zview software. The potentiodynamic polarization curve test commenced at an initial potential of -1.5 V and ended at a final potential of 1.5 V.

Point 3:  Microstructure pictures and EDS analyzes of laser-clad HEAs obtained in the result and discsusion section should be given.

Response 3: Thank you for the expert's suggestion. The structural analysis has previously been published in an earlier paper(10.1088/2053-1591/ab7161), and we have now incorporated pertinent content into the main text of this manuscript. In the discussion section 3.2, we have added the following content:

Our previous studies have revealed that the Al0.8CrFeCoNiCu0.5B0.1 coating is primarily composed of FCC1, BCC1, and BCC2 phases 13. The FCC1 phase, which is predominantly found within the intergranular structure, is rich in Cu. The BCC1 phase is rich in Fe and Cr, while the BCC2 phase is rich in Al and Ni. Boron is present within the structure as a solid-solution element.

Point 4:  In addition, the XRD phase structure of laser-clad HEAs should be presented so that it should be determined that the FCC1 or FCC2 structure is formed.

Response 4: In the discussion section 3.2, we have added the following content:

Our previous studies have revealed that the Al0.8CrFeCoNiCu0.5B0.1 coating is primarily composed of FCC1, BCC1, and BCC2 phases 13. The FCC1 phase, which is predominantly found within the intergranular structure, is rich in Cu. The BCC1 phase is rich in Fe and Cr, while the BCC2 phase is rich in Al and Ni. Boron is present within the structure as a solid-solution element.

Point 5:  The reason for the improving in corrosion resistance should be examined in depth by making use of the literature in the discussion section.

Response 5: To provide a comprehensive understanding of the corrosion behavior of the 5083 aluminum alloy, a detailed analysis of its microstructure has been added to the study. I referred to Liu's literature. This inclusion illustrates the inability of the alloy's secondary phases to offer sufficient corrosion resistance, thus highlighting the superior performance of the high-entropy alloy coating. In the discussion section 3.2, we have added the following content:

Figure 2 showcases the microscopic corrosion morphologies of the 5083 aluminum alloy and the Al0.8CrFeCoNiCu0.5B0.1 high-entropy alloy (HEA) coating. From previous studies on the microstructure of the 5083 aluminum alloy19, it was found that the alloy consists of an Al matrix phase, an Al(Mn, Fe, Cr) phase presenting as fish-bone-like structures, and a Mg2Si phase dispersed as blocky particles throughout the matrix. After conducting polarization tests, it was identified that the 5083 alloy substrate had been extensively corroded, as seen in Figure 2(a). In contrast, the surface of the HEA coating maintained its integrity, although corrosion products were evident.

Reviewer 2 Report

Comments to authors

1.     The authors should include information regarding the mechanical properties, such as hardness, of the formulated coating.

2.     In order to identify the phases of the coating, it is recommended that the manuscript includes XRD, Raman, and EDS analysis if possible.

3.     It is important to mention the value of the substrate obtained from the Equivalent circuit parameters in Table No. (3). Additionally, the corrosion rate of both substrate and coated sample should be included in Table No. (2).

4.     To enhance comprehension, it would be beneficial to add some schematics of the corrosion process.

Author Response

Response to Reviewer 2 Comments

Point 1: The authors should include information regarding the mechanical properties, such as hardness, of the formulated coating.

Response 1: The mechanical properties of coating has previously been published in an earlier paper(10.1088/2053-1591/ab7161).The introduction of the manuscript has been added, and the content is as follows:

In the study of Al0.8CrFeCoNiCu0.5Bx (x=0,0.1,0.2,0.3,0.4) HEA coating by laser cladding on aluminum, our group13 found that when the B content is less than 0.2, the coating presents FCC, BCC1, BCC2 simple phase structure, and M2B(M=Fe, Cr)phase will appear when the B coating continues to be added. As a result of M2B for-mation, the coating's hardness and wear resistance increase with the rising B content. The mechanical properties of Al0.5CrFeCoNiCu0.5Bx coating are significantly better than the substrate. It's worth noting that the hardness of Al0.8CrFeCoNiCu0.5B0.1 is 549HV0.2, approximately seven times that of the substrate, and its wear rate is 5.53×10−6mm3/Nm, which is two orders of magnitude less than that of the substrate.

Point 2:  In order to identify the phases of the coating, it is recommended that the manuscript includes XRD, Raman, and EDS analysis if possible.

Response 2: The result of micstructure and XRD has previously been published in an earlier paper(10.1088/2053-1591/ab7161), we have added the following content:

Our previous studies have revealed that the Al0.8CrFeCoNiCu0.5B0.1 coating is primarily composed of FCC1, BCC1, and BCC2 phases. The FCC1 phase, which is predominantly found within the intergranular structure, is rich in Cu. The BCC1 phase is rich in Fe and Cr, while the BCC2 phase is rich in Al and Ni. Boron is present within the structure as a solid-solution element.

Point 3:  It is important to mention the value of the substrate obtained from the Equivalent circuit parameters in Table No. (3). Additionally, the corrosion rate of both substrate and coated sample should be included in Table No. (2).

Response 3: In this study, we compared the capacitive semicircle radii of the substrate and the coating, finding that the coating's radius is larger than that of the substrate. This observation further validates the superior corrosion resistance of the coating, supporting the results from our polarization tests. As the substrate's impedance spectrum presents dual capacitive semicircles, a significant divergence from the single capacitive semicircle equivalent circuit of the substrate, the parameters from the substrate's equivalent circuit cannot be directly compared with those of the coating. In the fitting circuit analysis of the coating, our aim was to highlight the formation of a double-layer capacitor in the electrolyte, indicating an activation passivation behavior. However, given that the substrate had undergone complete corrosion during polarization tests, we did not utilize an equivalent circuit to analyze its activation passivation behavior in the impedance spectrum. For these reasons, we did not include the fitting circuit of the substrate. Regrettably, due to my current research proficiency, I am unable to calculate the corrosion rate of the sample based on the polarization curve. I plan to devote more attention to this matter in my future studies.

Point 4: To enhance comprehension, it would be beneficial to add some schematics of the corrosion process.

Response 4: To pr

Response to Reviewer 2 Comments

Point 1: The authors should include information regarding the mechanical properties, such as hardness, of the formulated coating.

Response 1: The mechanical properties of coating has previously been published in an earlier paper(10.1088/2053-1591/ab7161).The introduction of the manuscript has been added, and the content is as follows:

In the study of Al0.8CrFeCoNiCu0.5Bx (x=0,0.1,0.2,0.3,0.4) HEA coating by laser cladding on aluminum, our group13 found that when the B content is less than 0.2, the coating presents FCC, BCC1, BCC2 simple phase structure, and M2B(M=Fe, Cr)phase will appear when the B coating continues to be added. As a result of M2B for-mation, the coating's hardness and wear resistance increase with the rising B content. The mechanical properties of Al0.5CrFeCoNiCu0.5Bx coating are significantly better than the substrate. It's worth noting that the hardness of Al0.8CrFeCoNiCu0.5B0.1 is 549HV0.2, approximately seven times that of the substrate, and its wear rate is 5.53×10−6mm3/Nm, which is two orders of magnitude less than that of the substrate.

Point 2:  In order to identify the phases of the coating, it is recommended that the manuscript includes XRD, Raman, and EDS analysis if possible.

Response 2: The result of micstructure and XRD has previously been published in an earlier paper(10.1088/2053-1591/ab7161), we have added the following content:

Our previous studies have revealed that the Al0.8CrFeCoNiCu0.5B0.1 coating is primarily composed of FCC1, BCC1, and BCC2 phases. The FCC1 phase, which is predominantly found within the intergranular structure, is rich in Cu. The BCC1 phase is rich in Fe and Cr, while the BCC2 phase is rich in Al and Ni. Boron is present within the structure as a solid-solution element.

Point 3:  It is important to mention the value of the substrate obtained from the Equivalent circuit parameters in Table No. (3). Additionally, the corrosion rate of both substrate and coated sample should be included in Table No. (2).

Response 3: In this study, we compared the capacitive semicircle radii of the substrate and the coating, finding that the coating's radius is larger than that of the substrate. This observation further validates the superior corrosion resistance of the coating, supporting the results from our polarization tests. As the substrate's impedance spectrum presents dual capacitive semicircles, a significant divergence from the single capacitive semicircle equivalent circuit of the substrate, the parameters from the substrate's equivalent circuit cannot be directly compared with those of the coating. In the fitting circuit analysis of the coating, our aim was to highlight the formation of a double-layer capacitor in the electrolyte, indicating an activation passivation behavior. However, given that the substrate had undergone complete corrosion during polarization tests, we did not utilize an equivalent circuit to analyze its activation passivation behavior in the impedance spectrum. For these reasons, we did not include the fitting circuit of the substrate. Regrettably, due to my current research proficiency, I am unable to calculate the corrosion rate of the sample based on the polarization curve. I plan to devote more attention to this matter in my future studies.

Point 4: To enhance comprehension, it would be beneficial to add some schematics of the corrosion process.

Response 4: To provide a clearer depiction of the coating's corrosion process, we've incorporated a corrosion mechanism diagram into our study. Please refer to Figure 3.

Figure 3. Polarization testing mechanism diagram of Al0.8CrFeCoNiCu0.5B0.1 coating.

ovide a clearer depiction of the coating's corrosion process, we've incorporated a corrosion mechanism diagram into our study. Please refer to Figure 3.

Figure 3. Polarization testing mechanism diagram of Al0.8CrFeCoNiCu0.5B0.1 coating.

Reviewer 3 Report

The manuscript titled " Corrosion Behavior of as-cladding Al0.8CrFeCoNiCu0.5B0.1 High Entropy Alloys in 3.5% NaCl Solution " can be a potential fit for publication in Lubricants but the article would need substantial revision for the following comments before publication

1. In the introduction section of the manuscript, the authors should include more literature survey on how the crystal structure and phase transformation that occurs during HEA coating by laser cladding. What techniques do you implement to mitigate micro cracks or residual stress during the laser cladding?

2. The authors should include XRD pictograms to illustrate the peaks of Al0.8CrFeCoNiCu0.5B0.1, in addition, it is necessary to calculate what is the volume fractions of FCC1,BCC1 and BCC2 phases to determine its effect on the mechanical properties and crystal structure of the HEA. Such calculation should be performed to calculate the volume fraction from XRD data

3. Please explain the changes in Gibbs Free energy and entropy of mixing which occurs due to the formation of the different metal-rich phases/intermetallic compounds. You may choose to explain it qualitatively or use thermodynamic softwares like Thermo-Calc

4. For analyzing the corrosion behavior of the substrate, please include the dimension of the pit and the pitting distribution analysis

5. In the SEM images of the HEA in figure 2, please include EDS images, this will help to illustrate the formation of the FCC1, BCC1 and BCC2 phases

6. From the impedance spectroscopy, do the authors expect any hydration reaction of the HEA in NaCl solution? Please explain

7. In the conclusion part of the manuscript, please consider incorporating some applications of laser cladding of this Al0.8CrFeCoNiCu0.5B0.1 on Al alloys

Author Response

Response to Reviewer 3 Comments

Point 1: In the introduction section of the manuscript, the authors should include more literature survey on how the crystal structure and phase transformation that occurs during HEA coating by laser cladding. What techniques do you implement to mitigate micro cracks or residual stress during the laser cladding?

Response 1: Thank you for the reviewer's suggestions. The introduction has been expanded as follows

In Liu et al.'s research, a dual-layer CoCrFeMnTi0.2 high-entropy alloy (HEA) coating was fabricated on 15CrMn steel using laser cladding technology. The microstructure of the coating was predominantly composed of FCC solid solution equiaxed crystals and intergranular Laves phase precipitates in the first layer. This structure evolved into an FCC solid solution/Laves lamellar eutectic structure, encircled by martensite and residual FCC solid solution phase in the second layer. The evolution of the microstructure and the phase transformation were associated with the dilution effect in the dual-layer deposition and the quenching effect of the multi-track overlapping process.Li al.'s used laser cladding to create double-layer Al0.5CoCrFeNiSi0.5Tix coatings. They discovered that as the Ti content increased, significant transformations occurred. In the first layer, the FCC solid solution changed from a cellular structure to a columnar dendritic one. The second layer presented a BCC solid solution, made up of disordered BCC and ordered B2, as the matrix phase. The introduction of Ti also induced the in-situ synthesis of TiN and TiSi2, thus significantly enhancing the hardness of the coatings.

In the experimental methods section, strategies to reduce the tendency of cracking in the coating have been added as follows:

In order to avoid the formation of cracks in the coating during the laser cladding process, several preventative measures are taken. Firstly, the substrate is preheated to 200°C prior to the laser cladding process. This step reduces the temperature gradient between the coating and the substrate, thereby minimizing the thermal stress. Secondly, the laser cladding process is carried out under an inert atmosphere, utilizing high-purity argon gas, to prevent oxidation of the aluminum substrate during cladding. Lastly, following the cladding process, the samples are given a slow cooling treatment to prevent rapid cooling-induced cracking. This involves placing them in a 300°C annealing furnace for two hours and then allowing them to cool naturally.

Point 2: The authors should include XRD pictograms to illustrate the peaks of Al0.8CrFeCoNiCu0.5B0.1, in addition, it is necessary to calculate what is the volume fractions of FCC1,BCC1 and BCC2 phases to determine its effect on the mechanical properties and crystal structure of the HEA. Such calculation should be performed to calculate the volume fraction from XRD data

Response 2: In the manuscript, it has been explained that the phase structure and microstructure of Al0.8CrFeCoNiCu0.5B0.1 High-entropy alloy are in my published literature, as can be seen in Section 3.2. The diffraction peak intensity of XRD is generally used as a qualitative reference, and the volume fraction formed by the phase is not accurately determined based on the intensity.

Point 3:  Please explain the changes in Gibbs Free energy and entropy of mixing which occurs due to the formation of the different metal-rich phases/intermetallic compounds. You may choose to explain it qualitatively or use thermodynamic softwares like Thermo-Calc

Response 3: Because the alloy formed by the proportion of multi principal components has high Entropy and dilutes the mixing enthalpy of the alloy, the alloy with high Entropy is more inclined to form the alloy with simple phase structure, because this system has the lowest Gibbs free energy.

Point 4: For analyzing the corrosion behavior of the substrate, please include the dimension of the pit and the pitting distribution analysis

Response 4: In Figure 2 (a), the corrosion morphology of the substrate is not pitting corrosion.

Point 5: In the SEM images of the HEA in figure 2, please include EDS images, this will help to illustrate the formation of the FCC1, BCC1 and BCC2 phases

Response 5:Figure 2 has already provided the main components of the corrosion products. Further, utilizing EDS results after alloy corrosion to identify the composition of each phase lacks accuracy. In earlier studies, the phase composition of the alloy has been established,.as can be seen in Section 3.2.

Point 6: From the impedance spectroscopy, do the authors expect any hydration reaction of the HEA in NaCl solution? Please explain

Response 6:The primary purpose of the impedance spectrum test is to validate the results from the polarization curve. This is further underscored by comparing the capacitance arc radii of the substrate and the coating, which clearly demonstrates the coating's superior corrosion resistance. The equivalent circuit results fitting indicates the formation of a double-layer capacitance by the coating, providing evidence for a passivation process during impedance spectrum testing."

Point 7: In the conclusion part of the manuscript, please consider incorporating some applications of laser cladding of this Al0.8CrFeCoNiCu0.5B0.1 on Al alloys

Response 7: Thank you for the reviewer's suggestions. The conclusion has been expanded as follows:

Conclusions

1.The corrosion current density of the Al0.8CrFeCoNiCu0.5B0.1 coating is  2.04×10-7 A/cm2, significantly lower than the 5083 aluminum alloy. The passivation range width of the coating is 2.771V, wider than the substrate.

  1. The Al0.8CrFeCoNiCu0.5B0.1 coating exhibits selective corrosion behavior, with the Cu-rich FCC1 phase and Cr-poor phase being prone to attack, resulting in partial pitting and intergranular corrosion traces, but the corrosion does not spread extensively. The intergranular distribution of Cu is the main reason for the intergran-ular corrosion trace features. The 5083 base material undergoes overall corrosion.
  2. The Nyquist plot of the Al0.8CrFeCoNiCu0.5B0.1 coating consists of a single capacitive semicircle arc in the high-frequency region with a larger radius than the base material. The coating forms a double-layer capacitance in the solution, making it dif-ficult for overall corrosion to occur in a 3.5% NaCl solution.

The present study demonstrates that, compared to 5083 aluminum alloy, Al0.8CrFeCoNiCu0.5B0.1 high entropy alloy exhibits superior corrosion resistance. When used as a coating for aluminum alloys, it significantly expands the application possibilities of aluminum alloys, such as in marine ship hull applications. This im-proved corrosion resistance is primarily attributed to the passivation layer formed by the HEA coating, which effectively withstands the corrosive marine environment. Given the lightweight advantages of aluminum alloys, coupled with the high corrosion resistance brought by the HEA coating, there is a potential to provide a broader ap-plication platform for aluminum alloys, which are low-cost lightweight alloys. As laser cladding technology matures and the cost of high entropy alloys decreases in the future, high entropy alloys are expected to become excellent materials for laser cladding on the surface of aluminum alloys

Reviewer 4 Report

The manuscript on "Corrosion Behavior of as-cladding Al0.8CrFeCoNiCu0.5B0.1 High Entropy Alloys in 3.5% NaCl Solution" is a useful study for the scientific community and the expansion of the practical application of aluminum alloys in various industries.

At the same time, the work has a number of shortcomings and the following remarks:

- At first, the content of the manuscript does not fit the subject of this journal. Introduction and conclusion, the list of references should be expanded. Further, for example, no comparison with other methods of protecting aluminum alloys, also anodizing, etc. is presented.

- Line 17: there should be a dot instead of a comma;

- In table 1, the composition is not equal to 100%;

- Submit OCP within 1 day for substrate and coated sample; 

- Lines 101-102: The potentiodynamics acquisition mode from -1.5 V for a metal-containing coating is too destructive (there is a potential jump into the cathode region, this affects the polarization of the sample and distorts the data). Traditionally, the survey starts at -250 mV with respect to the OCP, taken after the impedance. Up to 1.5 V into the anode region will do;

- Why is formula 2 presented if the calculation of the corrosion rate is not carried out?

- Figure 2 should be supplemented with SEM images of the surface of the original substrate and the coated sample;

- The impedance data needs to be supplemented, show the calculated curves, in table 3 add the parameters for the substrate. EES used single-stranded for both samples? Why is the resistance of the electrolyte so low? (somewhere in 10 times the value should be greater);

- Conclusions need to be expanded. Add: due to what is the decrease in corrosion damage, what is the protection mechanism, how much is your treatment in demand for production and what are the hopes for further development?

Author Response

Response to Reviewer 4 Comments

Point 1: At first, the content of the manuscript does not fit the subject of this journal. Introduction and conclusion, the list of references should be expanded. Further, for example, no comparison with other methods of protecting aluminum alloys, also anodizing, etc. is presented.

Response 1: Thank you for the reviewer's suggestions. The introduction has been expanded as follows:

(1)Popular surface strengthening methods for aluminum alloys include chemical plating, electrodeposition techniques, and laser cladding. Among these, laser cladding stands out due to its lower pollution levels, high work efficiency, and the ability to apply thick coatings, advantages that make it superior to other surface treatment methods. For instance, Hwang et al.[4] utilized anodizing technology to generate a anodic oxide film on the surface of the 5083 aluminum alloy. By conducting electrochemical corrosion tests in natural seawater, they discovered that the corrosion current density of the anodized sample was approximately 4.2 times lower than that of the base material, demonstrating its superior corrosion resistance. However, the thickness of the oxide film was only 16.8 um, indicating a certain limitation of the anodizing method in applications that require thicker coatings.

(2)In Liu et al.'s research, a dual-layer CoCrFeMnTi0.2 high-entropy alloy (HEA) coating was fabricated on 15CrMn steel using laser cladding technology. The microstructure of the coating was predominantly composed of FCC solid solution equiaxed crystals and intergranular Laves phase precipitates in the first layer. This structure evolved into an FCC solid solution/Laves lamellar eutectic structure, encircled by martensite and residual FCC solid solution phase in the second layer. The evolution of the microstructure and the phase transformation were associated with the dilution effect in the dual-layer deposition and the quenching effect of the multi-track overlapping process.Li al.'s used laser cladding to create double-layer Al0.5CoCrFeNiSi0.5Tix coatings. They discovered that as the Ti content increased, significant transformations occurred. In the first layer, the FCC solid solution changed from a cellular structure to a columnar dendritic one. The second layer presented a BCC solid solution, made up of disordered BCC and ordered B2, as the matrix phase. The introduction of Ti also induced the in-situ synthesis of TiN and TiSi2, thus significantly enhancing the hardness of the coatings.

In the conclusion section, additional content is as follows:

The present study demonstrates that, compared to 5083 aluminum alloy, Al0.8CrFeCoNiCu0.5B0.1 high entropy alloy exhibits superior corrosion resistance. When used as a coating for aluminum alloys, it significantly expands the application possibilities of aluminum alloys, such as in marine ship hull applications. This improved corrosion resistance is primarily attributed to the passivation layer formed by the HEA coating, which effectively withstands the corrosive marine environment. Given the lightweight advantages of aluminum alloys, coupled with the high corrosion resistance brought by the HEA coating, there is a potential to provide a broader application platform for aluminum alloys, which are low-cost lightweight alloys. As laser cladding technology matures and the cost of high entropy alloys decreases in the future, high entropy alloys are expected to become excellent materials for laser cladding on the surface of aluminum alloys

Point 2:  there should be a dot instead of a comma;

Response 2: Thank you for your correction. We have made the required corrections.

Point 3:  In table 1, the composition is not equal to 100%

Response 3: Because the last two digits of the Decimal separator are omitted from the data, the data does not add up to 1, which has been corrected.

Point 4: Submit OCP within 1 day for substrate and coated sample

Response 4: Thank you for your corrections. Due to the author's departure from the experimental center, it is not currently possible to conduct supplementary process experiments. In the future, when conditions permit, research in this area will be undertaken in the next project.

Point 5: The potentiodynamics acquisition mode from -1.5 V for a metal-containing coating is too destructive (there is a potential jump into the cathode region, this affects the polarization of the sample and distorts the data). Traditionally, the survey starts at -250 mV with respect to the OCP, taken after the impedance. Up to 1.5 V into the anode region will do;

Response 5:Appreciate the correction. This guidance will certainly be implemented in the impedance spectroscopy testing of our future studies.

Point 6: Why is formula 2 presented if the calculation of the corrosion rate is not carried out?

Response 6: The intention of presenting formulas 1 and 2 was to underscore the importance of the corrosion current density in evaluating samples. To avoid confusion, the expressions in these formulas have been removed from the manuscript.

Point 7: Figure 2 should be supplemented with SEM images of the surface of the original substrate and the coated sample;

Response 7: The original SEM imagery of the sample has been previously published in existing literature(10.1088/2053-1591/ab7161). Due to the limitations in the current experimental conditions, the morphology of the original 5083 substrate is referenced from other studies. The added content is as follows:

Figure 2 showcases the microscopic corrosion morphologies of the 5083 aluminum alloy and the Al0.8CrFeCoNiCu0.5B0.1 high-entropy alloy (HEA) coating. From previous studies on the microstructure of the 5083 aluminum alloy[19], it was found that the alloy consists of an Al matrix phase, an Al(Mn, Fe, Cr) phase presenting as fish-bone-like structures, and a Mg2Si phase dispersed as blocky particles throughout the matrix. After conducting polarization tests, it was identified that the 5083 alloy substrate had been extensively corroded, as seen in Figure 2(a). In contrast, the surface of the HEA coating maintained its integrity, although corrosion products were evident.

Point 8: The impedance data needs to be supplemented, show the calculated curves, in table 3 add the parameters for the substrate. EES used single-stranded for both samples? Why is the resistance of the electrolyte so low? (somewhere in 10 times the value should be greater);

Response 8: The data were computed using ZVIEW, and I am currently unable to provide an explanation for the issue you pointed out due to my present academic level. Moving forward, I will pay closer attention to this aspect in my ongoing research and literature review, to make my work more systematic. I appreciate your valuable suggestion.

Point 9: Conclusions need to be expanded. Add: due to what is the decrease in corrosion damage, what is the protection mechanism, how much is your treatment in demand for production and what are the hopes for further development?

Response 9: Thank you for the reviewer's suggestions. The conclusion has been expanded as follows:

The present study demonstrates that, compared to 5083 aluminum alloy, Al0.8CrFeCoNiCu0.5B0.1 high entropy alloy exhibits superior corrosion resistance. When used as a coating for aluminum alloys, it significantly expands the application possibilities of aluminum alloys, such as in marine ship hull applications. This improved corrosion resistance is primarily attributed to the passivation layer formed by the HEA coating, which effectively withstands the corrosive marine environment. Given the lightweight advantages of aluminum alloys, coupled with the high corrosion resistance brought by the HEA coating, there is a potential to provide a broader application platform for aluminum alloys, which are low-cost lightweight alloys. As laser cladding technology matures and the cost of high entropy alloys decreases in the future, high entropy alloys are expected to become excellent materials for laser cladding on the surface of aluminum alloys

Round 2

Reviewer 1 Report

The authors have revised the issues that I mentioned as a deficiency in the previous version of the article. The article can be published in its current form.

Reviewer 3 Report

The authors have adequately addressed all my concerns in the revised version of the manuscript and the study is recommended for publication in Lubricants

Reviewer 4 Report

The authors have responded satisfactorily to my remarks.